# Comparisons of Outcomes between Patients with Direct and Indirect Acute Respiratory Distress Syndrome Receiving Extracorporeal Membrane Oxygenation

**DOI:** 10.3390/membranes11080644

**Published:** 2021-08-22

**Authors:** Li-Chung Chiu, Li-Pang Chuang, Shih-Wei Lin, Hsin-Hsien Li, Shaw-Woei Leu, Ko-Wei Chang, Chi-Hsien Huang, Tzu-Hsuan Chiu, Huang-Pin Wu, Feng-Chun Tsai, Chung-Chi Huang, Han-Chung Hu, Kuo-Chin Kao

**Affiliations:** 1Department of Thoracic Medicine, Chang Gung Memorial Hospital, Chang Gung University College of Medicine, Taoyuan 33305, Taiwan; pomd54@cgmh.org.tw (L.-C.C.); r5243@adm.cgmh.org.tw (L.-P.C.); ec108146@adm.cgmh.org.tw (S.-W.L.); swleu@cgmh.org.tw (S.-W.L.); b9302072@cgmh.org.tw (K.-W.C.); b9502072@cgmh.org.tw (C.-H.H.); kodochasana@gmail.com (T.-H.C.); cch4848@cgmh.org.tw (C.-C.H.); kck0502@cgmh.org.tw (K.-C.K.); 2Graduate Institute of Clinical Medical Sciences, College of Medicine, Chang Gung University, Taoyuan 33302, Taiwan; 3Department of Thoracic Medicine, New Taipei Municipal TuCheng Hospital and Chang Gung University, Taoyuan 33302, Taiwan; 4Institute of Emergency and Critical Care Medicine, School of Medicine, National Yang Ming Chiao Tung University, Taipei 11221, Taiwan; hsinhsien@mail.cgu.edu.tw; 5Department of Respiratory Therapy, Chang Gung University College of Medicine, Taoyuan 33302, Taiwan; 6Division of Pulmonary, Critical Care and Sleep Medicine, Chang Gung Memorial Hospital, Keelung 20401, Taiwan; whanpyng@cgmh.org.tw; 7Division of Cardiovascular Surgery, Chang Gung Memorial Hospital, Taoyuan 33305, Taiwan; lutony@cgmh.org.tw; 8Department of Respiratory Therapy, Chang Gung Memorial Hospital, Chang Gung University College of Medicine, Taoyuan 33305, Taiwan

**Keywords:** acute respiratory distress syndrome, extracorporeal membrane oxygenation, direct lung injury, indirect lung injury, mortality

## Abstract

Acute respiratory distress syndrome (ARDS) is a heterogeneous syndrome caused by direct (local damage to lung parenchyma) or indirect lung injury (insults from extrapulmonary sites with acute systemic inflammatory response), the clinical and biological complexity can have a profound effect on clinical outcomes. We performed a retrospective analysis of 152 severe ARDS patients receiving extracorporeal membrane oxygenation (ECMO). Our objective was to assess the differences in clinical characteristics and outcomes of direct and indirect ARDS patients receiving ECMO. Overall hospital mortality was 53.3%. A total of 118 patients were assigned to the direct ARDS group, and 34 patients were assigned to the indirect ARDS group. The 28-, 60-, and 90-day hospital mortality rates were significantly higher among indirect ARDS patients (all *p* < 0.05). Cox regression models demonstrated that among direct ARDS patients, diabetes mellitus, immunocompromised status, ARDS duration before ECMO, and SOFA score during the first 3 days of ECMO were independently associated with mortality. In indirect ARDS patients, SOFA score and dynamic compliance during the first 3 days of ECMO were independently associated with mortality. Our findings revealed that among patients receiving ECMO, direct and indirect subphenotypes of ARDS have distinct clinical outcomes and different predictors for mortality.

## 1. Introduction

The clinical and biological heterogeneity of acute respiratory distress syndrome (ARDS) involves complex pathophysiologic mechanisms encompassing a multitude of risk factors, all of which can contribute to distinct clinical outcomes and the varied responses to therapeutics observed in failed clinical trials [1,2,3,4,5,6,7].

The main priority in caring for patients with ARDS is identifying and treating the underlying etiologies, which can be divided into those directly or indirectly related to lung injury [1]. Direct (primary or pulmonary) ARDS results from an insult that directly affects lung parenchyma (e.g., pneumonia, aspiration of gastric contents), and indirect (secondary or extrapulmonary) ARDS results from an insult outside of the lungs with an acute systemic inflammatory response (e.g., nonpulmonary sepsis, trauma, pancreatitis) [3]. Separating ARDS patients into homogenous subphenotypes (e.g., direct and indirect ARDS) could have clinical implications affecting the means by which clinical trials are conducted, and identification of ARDS subphenotypes may enable the aspiration of precision medicine for ARDS [7,8].

Alveolar epithelial injury with a local alveolar inflammatory response is a direct insult resulting from pulmonary ARDS (i.e., direct or primary ARDS) with predominant consolidation, whereas systemic vascular endothelial damage caused by inflammatory mediators in the bloodstream is the indirect insult of extrapulmonary ARDS (i.e., indirect or secondary ARDS) with prevalent interstitial edema, ground-glass opacification, and alveolar collapse [2,3,5,6,7]. Irrespective of the initial insult, the final result is a disruption to the pulmonary alveolar-capillary barrier with consequent hypoxemia, inflammation, noncardiogenic pulmonary edema, and eventual organ failure [9].

Previous studies have reported that patients with direct or indirect ARDS differ in terms of pathophysiology, biochemistry, radiography, respiratory mechanics, and responses to ventilatory and clinical management strategies, all of which contribute to diverse clinical outcomes and mortality [2,3,4,5,6,10,11]. Extracorporeal membrane oxygenation (ECMO) is considered to be a rescue therapy for refractory hypoxemia in cases of severe ARDS [12,13,14]. Differences in the etiology of ARDS (i.e., direct or indirect lung injury) may also be responsible for the observed diversity of clinical characteristics and clinical outcomes. Note, however, that few published reports have addressed this issue.

Our objective in this study was to examine correlations among clinical and ventilatory variables, clinical outcomes and mortality, and predictors of hospital mortality in patients with direct or indirect ARDS undergoing ECMO.

## 2. Materials and Methods

### 2.1. Study Design and Patient Inclusion

This retrospective study enrolled patients undergoing ECMO for severe ARDS between May 2006 and October 2015 in the medical and surgical ICUs at a tertiary care referral center, Chang Gung Memorial Hospital (CGMH) in Taiwan, with a 3700-bed general ward and a 278-bed adult ICU. Patients who were younger than 20 years, had cancer with a life expectancy of less than 5 years, had significant comorbidities or multiple organ failure refractory to therapy (i.e., a moribund condition), or had died within 3 days after ECMO, were excluded from analysis. At our institution, the decision to initiate ECMO cannulation is made by the treating intensivist and cardiac surgeon. Criteria for ECMO initiation in severe ARDS patients was indicated if the PaO_2_/FiO_2_ ratio was less than 80 mm Hg for more than 6 h when conventional lung-protective ventilation with higher airway pressures fails. All severe ARDS patients receiving ECMO support were deeply sedated and paralyzed during the initial phase of ECMO support. Mechanical ventilator settings were collected during the neuromuscular blockade. The local Institutional Review Board for Human Research approved this study (CGMH IRB No. 201600632B0), and the need for informed consent was waived due to the retrospective nature of the study.

### 2.2. Definitions

ARDS was defined in accordance with the Berlin criteria [15]. Causes of ARDS were determined by the treating intensivists. The classification of ARDS into direct or indirect ARDS was made independently by two investigators (L.-C.C. and L.-P.C.) in a blinded manner according to the causes of ARDS derived from medical charts. Any discrepancies were revised by the two investigators to reach a final consensus, both of whom agreed with the final classifications. Mechanical power was calculated using the following Equation [16]:

Mechanical power (Joules/minutes) (J/min) = 0.098 × tidal volume × respiratory rate × (peak inspiratory pressure − 1/2 × driving pressure).

Ventilatory ratio was calculated as [minute ventilation (mL/min) × PaCO_2_ (mm Hg)]/(predicted body weight × 100 × 37.5) [17].

The cumulative fluid balance was defined as cumulative total fluid input minus cumulative total fluid output.

### 2.3. Data Collection

Demographic data, etiologies of ARDS (direct or indirect lung injury), comorbidities, laboratory, and clinical data were recorded. Organ dysfunction was assessed by the Sequential Organ Failure Assessment (SOFA) scores, which were calculated before ECMO initiation and on days 1, 2, and 3 after ECMO support. All changes in arterial blood gas and ventilator setting variables were obtained before ECMO initiation and every day, and the values on days 1, 2, and 3 were analyzed. The duration of ECMO, duration of mechanical ventilation, the length of ICU stay, the length of hospital stay, and survival status, were evaluated.

### 2.4. Statistical Analysis

Statistical analysis was performed with SPSS Statistics version 26.0. Descriptive statistics were used to describe patient characteristics. Mean and standard deviation were computed for normally distributed continuous variables, whereas continuous variables not normally distributed were presented as a median and interquartile range. Student *t* test was performed for comparison of normally distributed data, and a Mann–Whitney *U* test was used for nonparametric data. Categorical variables were presented as frequencies and percentages, and were compared by the chi-square test for equal proportions, or the Fisher’s exact test. Univariate analysis was used to identify risk factors associated with hospital mortality in the direct and indirect ARDS subgroups first, and then a Cox proportional hazard regression model with stepwise selection was constructed. The results were presented as hazard ratios (HR) with 95% confidence intervals (CI). The probability of survival was analyzed with the use of the Kaplan–Meier method and compared between groups with the use of the log-rank test. Statistical significance was considered when a two-sided *p* value was less than 0.05.

## 3. Results

### 3.1. Study Patients

During the study period, a total of 189 patients with severe respiratory failure receiving ECMO were included. After excluding 37 patients, a total of 152 patients with severe ARDS rescued by ECMO were enrolled in the current analysis, and the overall hospital mortality rate was 53.3%. A total of 118 patients were assigned to the direct ARDS group, and 34 patients were assigned to the indirect ARDS group. Bacterial pneumonia (*n* = 55, 46.6%) was the primary risk factor for direct ARDS, whereas nonpulmonary sepsis (*n* = 20, 58.8%) was the primary risk factor for indirect ARDS. Compared to mortality in the overall patient population (53.3%), hospital mortality was higher for bacterial pneumonia and nonpulmonary sepsis (65.5% and 75%, respectively), whereas hospital mortality for influenza pneumonia was lower (40.9%) (Figure 1).

### 3.2. Comparisons of Direct and Indirect ARDS Patients

As shown in Table 1, no significant differences were observed between direct and indirect ARDS patients in terms of age, gender, body mass index, or major comorbidities. Prior to ECMO support, we observed no significant differences in terms of ARDS severity (i.e., PaO_2_/FiO_2_); however, direct ARDS patients presented a significantly higher lung injury score. In terms of ventilator settings prior to ECMO, mechanical power and minute ventilation were significantly higher in direct ARDS patients (both *p* < 0.05). The median ARDS duration prior to ECMO implantation was 28 h for all patients, with no significant difference between the direct and indirect ARDS groups. Indirect ARDS patients received venoarterial (VA) ECMO implantation more often than direct ARDS patients did.

After ECMO initiation, there were no significant differences between the two groups in terms of ventilator settings, except for higher PEEP in direct ARDS patients. During the first 3 days of ECMO, SOFA scores and cumulative fluid balance were significantly higher in the indirect ARDS group (both *p* < 0.05).

### 3.3. Clinical Outcomes of Direct and Indirect ARDS Patients

As shown in Table 2, 28-, 60-, and 90-day hospital mortality rates were significantly higher among indirect ARDS patients than among those with direct ARDS (all *p* < 0.05). Overall, the duration of ECMO support was higher among patients with direct ARDS. We observed no significant differences between the two groups in terms of the duration of mechanical ventilation, the length of ICU stay, the length of hospital stay, 28-day ECMO-free days, nor 28-day or 60-day ventilator-free days. Kaplan–Meier estimates revealed a significant difference in 90-day survival between patients with direct and indirect ARDS (52.5% vs 38.2%, respectively; *p* = 0.041, log-rank test) (Figure 2).

### 3.4. Comparison of Direct and Indirect ARDS Patients in Terms of Survival

In the direct ARDS group, survivors were younger than non-survivors and ARDS duration prior to ECMO was shorter. Furthermore, a higher percentage of survivors had diabetes mellitus, and a lower percentage of survivors were immunocompromised, compared with non-survivors (see Table 3). We observed no differences between survivors and non-survivors in terms of ventilator settings before or during ECMO. Survivors presented lower SOFA scores, lower cumulative fluid balance, and the incidence of using inotropes was lower during ECMO compared with non-survivors.

In the indirect ARDS group, survivors were younger than non-survivors, and a lower percentage was immunocompromised. None of the survivors had chronic liver disease or chronic kidney disease. ARDS duration before ECMO was shorter among survivors than among non-survivors, and SOFA scores before and during ECMO were lower (all *p* < 0.05). We observed no significant differences between the two groups in terms of ventilator settings prior to ECMO support; however, mechanical power and peak inspiratory pressure were significantly lower among survivors compared with non-survivors, and dynamic compliance was higher among survivors after ECMO support (all *p* < 0.05).

### 3.5. Factors Associated with Hospital Mortality in Cases of Direct and Indirect ARDS

After adjusting for significant confounding variables, Cox proportional hazard regression models revealed that diabetes mellitus was an independent factor for decreased risk of death in direct ARDS patients, whereas immunocompromised status, ARDS duration before ECMO, and SOFA score during the first 3 days of ECMO were independently associated with an increased risk of death. In indirect ARDS patients, a higher SOFA score and a lower dynamic compliance during the first 3 days of ECMO were independently associated with higher hospital mortality (Table 4).

### 3.6. Comparisons of Direct and Indirect ARDS Patients after Excluding VA ECMO Patients

As shown in Table 5, there were no significant differences between direct and indirect ARDS patients in terms of age, gender, body mass index, or comorbidities. Before ECMO initiation, direct ARDS patients had significantly higher lung injury scores. In terms of ventilator settings prior to ECMO, no significant difference between the two groups were found.

After ECMO initiation, no significant differences were observed between the two groups in terms of ventilator settings, except for higher PEEP in direct ARDS patients. SOFA scores and cumulative fluid balance were significantly higher in indirect ARDS patients during the first 3 days of ECMO (both *p* < 0.05). The 28-, 60-, and 90-day hospital mortality rates were significantly higher among indirect ARDS patients than among direct ARDS patients (all *p* < 0.05).

## 4. Discussion

The primary insight in this research was the fact that 28-, 60-, and 90-day hospital mortality rates were higher among patients with indirect ARDS than among patients with direct ARDS. Furthermore, except for organ failure (i.e., SOFA scores), the two groups differed entirely in terms of independent predictors of hospital mortality.

The ability to identify and treat the underlying etiologies of ARDS is crucial to the effectiveness of ECMO. Studies have found that influenza pneumonia-induced ARDS is associated with better outcomes. Mortality rates tend to be higher in cases of ARDS induced by non-pulmonary sepsis than in cases induced by pneumonia [13,18]. The higher mortality in indirect (extrapulmonary) ARDS patients may be due to the complications of underlying diseases and the difficulties in treating these fatal complications [19]. Our study showed that one ARDS patient, due to trauma, had an intracranial hemorrhage and two ARDS patients due to acute pancreatitis had necrotizing pancreatitis, and these complications eventually contributed to mortality. Our findings were similar to those in the literature. Nonetheless, researchers have yet to comprehensively determine the differences between direct and indirect ARDS in terms of clinical features, ventilatory parameters before and after ECMO, clinical outcomes, and predictors for mortality.

One recent study reported that in the early stages after ARDS diagnosis (median time of 4 days), gas exchange impairment was significantly more pronounced in cases of pulmonary (i.e., direct) ARDS [4] than in cases of indirect lung injury. Pathophysiology and radiography results also revealed that in cases of direct lung injury, lung consolidation was more pronounced, making it less amenable to recruitment (i.e., stiffer lungs). This may contribute to poorer oxygenation with poorer responses to mechanical ventilation in cases of direct ARDS [2,3,5,7]. In the current study, the median ARDS duration prior to ECMO initiation was 28 h (i.e., early stage of disease). Compared with cases of indirect ARDS, patients with direct ARDS presented significantly higher lung injury scores before ECMO initiation and received significantly higher mechanical power and insignificantly higher peak inspiratory pressure and higher mean airway pressure. These findings indicate that patients with direct ARDS may require higher airway pressures prior to ECMO in order to improve oxygenation, due perhaps to more lung consolidation (i.e., reduced recruitability).

ECMO facilitates ultra-protective ventilation to allow lower energy loads (i.e., mechanical power) and airway pressures, thereby mitigating ventilator-induced lung injury (VILI) and improving gas exchange [12,14,16]. It has been reported that there are survival benefits to severe ARDS patients receiving higher airway pressures to receive ECMO treatment [20]. After ECMO initiation for lung rest, we observed no significant differences between direct and indirect subgroups in terms of ventilator settings, except for higher PEEP in direct ARDS patients. Mechanical ventilator settings during ECMO were associated with mortality in severe ARDS patients [16,21,22], and lower lung compliance during ECMO was related to increased mortality [23]. Lower dynamic compliance was also independently associated with an increased hazard of death in indirect ARDS patients.

Previous studies have reported that patients with indirect ARDS face an elevated risk of hemodynamic impairment or shock, and a higher proportion of these patients receive vasopressors [2,6,11]. Sepsis is the main risk factor for indirect ARDS [1]. Latent class analysis of the distinct subphenotypes of ARDS revealed a stronger correlation between sepsis-associated ARDS and hyperinflammatory subphenotypes, as characterized by higher plasma concentrations of inflammatory biomarkers, a higher prevalence of vasopressor use, fewer organ failure-free days, and higher mortality [24]. VA ECMO is less frequently applied in cases of severe ARDS with refractory hypoxemia, except in cases of significant cardiac dysfunction requiring hemodynamic support. Based on all cases of ARDS supported using Extracorporeal Life Support Organization Registry, mortality rates were significantly higher among VA ECMO patients than among VV ECMO patients [25,26]. We found that VA ECMO was far more prevalent among patients suffering from indirect ARDS, due perhaps to a higher incidence of severe sepsis with a higher severity of illness (i.e., significantly higher SOFA scores and higher cumulative fluid status during the first 3 days of ECMO), often with considerable hemodynamic compromise, which may have contributed to the higher mortality in indirect subgroup. However, after excluding VA ECMO-supported patients, the 28-, 60-, and 90-day hospital mortality rates were still significantly higher among patients with indirect ARDS.

During the early phase of ECMO, positive fluid balance was independently associated with mortality [27]. Excess fluid accumulation may exacerbate tissue edema, stretch the vascular wall, worsen vascular permeability, and ultimately develop organ dysfunction with corresponding effects on clinical outcomes and mortality [28]. Although the causal relationship between fluid overload and organ dysfunction was difficult to determine due to the retrospective nature of our study, our findings showed that indirect ARDS patients had a significantly higher cumulative fluid balance and higher organ failure during early phase of ECMO than direct ARDS patients, which may contribute to higher mortality.

The most common cause of death among ARDS patients is multiorgan failure [1]. One previous study reported that among both direct and indirect ARDS patients, the number of organ failures was independently associated with mortality. They also reported that cases of organ failure were significantly more common among indirect ARDS patients than among direct ARDS patients [11]. Pneumonia and sepsis are the primary etiologies of direct and indirect ARDS, and sepsis is more commonly associated with more severe multiple organ dysfunction [1,4,6,10,24]. One international study reported that in severe ARDS patients, extrapulmonary organ failure during ECMO had a significantly negative impact on mortality [29]. In our study, we also found that sepsis was the main cause of indirect ARDS, and SOFA scores before ECMO were higher in indirect ARDS patients compared with direct ARDS patients; however, the difference was not significant. SOFA scores subsequently decreased during the first 3 days of ECMO in both direct and indirect ARDS patients, indicating that ECMO could facilitate a further reduction in ventilator load (i.e., mechanical power) to alleviate VILI by reducing the proinflammatory biotrauma response, thereby preventing further multi-organ failure [12,16,20,30]. However, SOFA scores during the first 3 days of ECMO remained significantly higher in patients with indirect ARDS, which may have contributed to the higher mortality in that group. Cox regression models revealed that in both groups, SOFA scores during the first 3 days of ECMO were independently associated with hospital mortality.

Other predictors for hospital mortality have been reported for direct and indirect ARDS. One study reported that diabetes mellitus is associated with a reduced risk of developing ARDS [31]; however, another study failed to detect any association between diabetes mellitus and the presence of ARDS, the risk of developing ARDS, clinical outcomes, or mortality in ARDS patients [32]. The mechanisms correlating diabetes and ARDS development remain unclear; however, researchers have posited the attenuation of cytokine release and impairment of neutrophil function as potential candidates [31]. One retrospective cohort study reported that preexisting diabetes mellitus was independently associated with a reduced risk of mortality only in direct ARDS patients, and diabetes was not found as a protective factor in indirect ARDS patients [11]. The impact of diabetes on severe ARDS patients treated with ECMO will require further research [13]. In the current study, we found that in direct ARDS patients receiving ECMO, diabetes mellitus did indeed have an impact on survival. Diabetes as a protective factor in indirect ARDS patients undergoing ECMO was not found in our study. However, one recent experimental rat model demonstrated that diabetes promoted proinflammatory cytokine release, renal damage, and pulmonary edema during ECMO [33]. Associations have been found between immunocompromised status and higher mortality in severe ARDS patients undergoing ECMO [13,18,29]. The optimal timing of ECMO initiation for severe ARDS has not been clearly defined; however, a longer ARDS duration prior to ECMO is associated with higher mortality [13,18,21,29]. We found that in direct ARDS patients, immunocompromised status and ARDS duration before ECMO were independently associated with higher mortality.

This study was hindered by a number of limitations. First, the relatively small number of indirect ARDS patients, the retrospective nature of our analysis, the fact that external validation was not performed, and the fact that all patients were from the long enrollment period of the previous years (2006–2015) and not recent years in a single tertiary care referral center. Second, the causes of ARDS may be multifactorial, and any number of enrolled patients may have had direct and indirect insults to ARDS. Furthermore, the fact that we did not focus on specific etiologies within the subgroups may have affected the results. Third, this study focused only on ARDS subphenotypes of direct or indirect lung injury. We did not analyze biological markers (e.g., inflammatory cytokines or biomarkers of lung epithelial injury or endothelial injury), radiography results (although ARDS patients receiving ECMO may preclude widespread clinical use of computed tomography scans), or respiratory mechanics. At this point, we are unsure whether identifying subphenotypes in terms of other clinical characteristics or combining biological profiles would have a better predictive value and explore more and new markers to follow up disease may need further research in the future. Fourth, although a recent meta-analysis demonstrated that corticosteroids treatment might reduce overall mortality and duration of mechanical ventilation in ARDS patients [34], they did not conduct subgroup analyses such as the underlying etiology of ARDS (i.e., direct or indirect injury). Corticosteroids use in ARDS remains highly controversial due to unclear benefits and the optimal dose and duration are unknown, and our study did not evaluate the possible impact of corticosteroids on clinical outcomes in ARDS patients receiving ECMO. Finally, our objective in this observational study was to investigate differences in clinical characteristics and outcomes between direct and indirect ARDS patients, without considering issues pertaining to causality. Thus, our results should be interpreted with care.

## 5. Conclusions

This study revealed that hospital mortality was significantly higher among patients with indirect ARDS receiving ECMO than among patients with direct ARDS. We also found that the two subgroups differed in terms of predictors for mortality. Future clinical trials of ECMO in severe ARDS patients could further stratify the different subgroups of ARDS patients based on merging clinical, radiographic, or biological features to reduce heterogeneity with the aim of developing potential prognostic indicators and therapeutic strategies to improve outcomes.

## Figures and Tables

**Figure 1 membranes-11-00644-f001:**
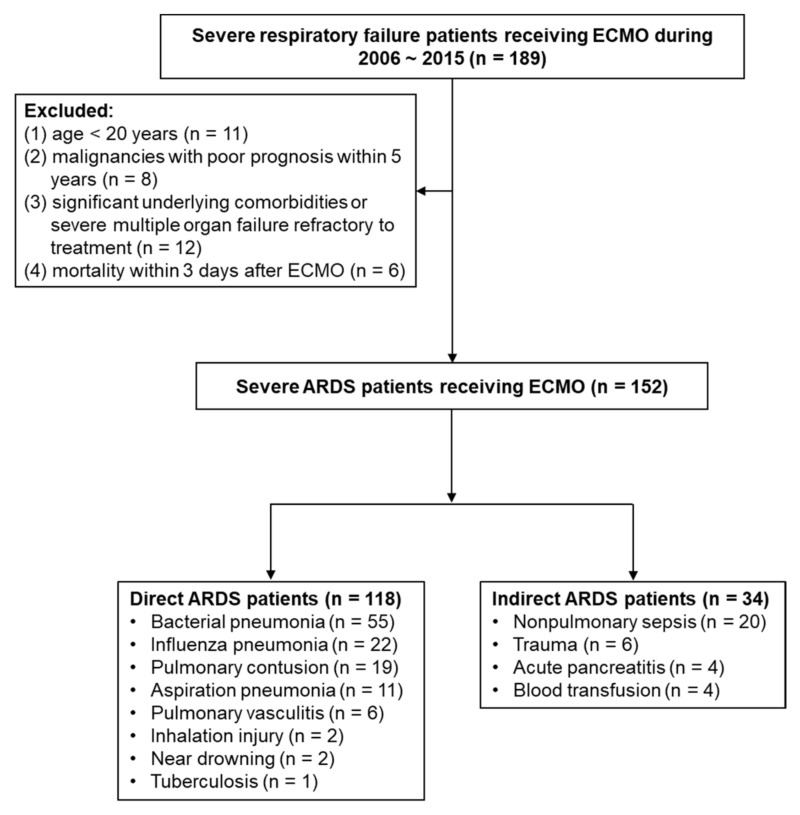
Flow diagram of enrolling patients with severe ARDS with ECMO support. (ARDS, acute respiratory distress syndrome; ECMO, extracorporeal membrane oxygenation).

**Figure 2 membranes-11-00644-f002:**
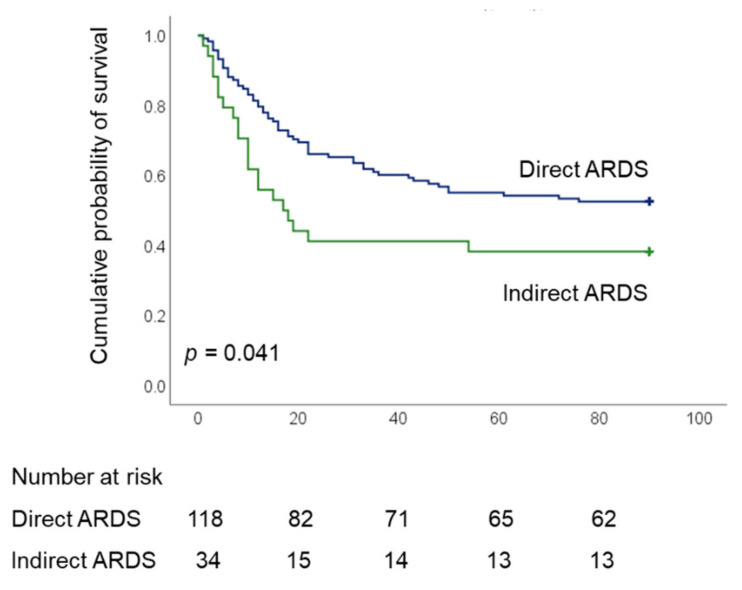
Kaplan–Meier 90-d survival curves of patients undergoing ECMO for severe acute respiratory distress syndrome, as stratified by direct and indirect ARDS (ARDS, acute respiratory distress syndrome; ECMO, extracorporeal membrane oxygenation).

**Table 1 membranes-11-00644-t001:** Background characteristics and clinical variables: Direct and indirect ARDS patients.

Variables	All	Direct ARDS	Indirect ARDS	*p*
	(*n* = 152)	(*n* = 118)	(*n* = 34)	
Age (years)	50.3 ± 16.4	50.8 ± 16.3	48.5 ± 16.6	0.463
Gender (male)	103 (67.8%)	83 (70.3%)	20 (58.8%)	0.206
Body mass index (kg/m^2^)	25.8 ± 5.3	25.9 ± 5.4	25.5 ± 4.8	0.766
Diabetes mellitus	40 (26%)	33 (28%)	7 (20.6%)	0.389
Chronic liver disease	21 (14%)	16 (13.6%)	5 (14.7%)	0.864
Immunocompromised status	40 (26%)	28 (23.7%)	12 (35.3%)	0.177
Chronic kidney disease	17 (11.2%)	13 (11%)	4 (11.8%)	1.000
SOFA score before ECMO	10.8 ± 3.2	10.6 ± 3.0	11.5 ± 3.4	0.154
Lung injury score before ECMO	3.4 ± 0.4	3.4 ± 0.4	3.3 ± 0.4	0.036
ARDS duration before ECMO (h)	28 (7–122)	29 (7–122)	24 (5–192)	0.306
pH before ECMO	7.28 ± 0.14	7.28 ± 0.14	7.26 ± 0.13	0.494
PaCO_2_ (mm Hg) before ECMO	52.5 ± 19.0	52.6 ± 19.2	51.9 ± 18.6	0.848
PaO_2_/FiO_2_ (mm Hg) before ECMO	63 (52–88)	63 (52–87)	65 (53–97)	0.430
Ventilator settings before ECMO				
Mechanical power (J/min)	23.8 ± 9.6	24.7 ± 9.5	20.5 ± 9.4	0.023
Tidal volume (mL/kg PBW)	7.7 ± 2.4	7.7 ± 2.3	7.7 ± 2.9	0.981
PEEP (cm H_2_O)	12.0 ± 2.8	12.2 ± 2.8	11.3 ± 2.8	0.106
Peak inspiratory pressure (cm H_2_O)	33.9 ± 6.5	34.2 ± 6.5	33.2 ± 6.6	0.457
Mean airway pressure (cm H_2_O)	18.6 ± 4.4	19.0 ± 4.4	17.5 ± 4.4	0.094
Dynamic compliance (mL/cm H_2_O)	22.6 ± 11.3	22.8 ± 11.2	21.8 ± 11.8	0.680
Minute ventilation (L/min)	10.6 ± 3.8	11.0 ± 3.9	9.2 ± 3.3	0.019
Ventilatory ratio	2.5 ± 1.1	2.5 ± 1.0	2.3 ± 1.2	0.183
ECMO venoarterial mode	24 (15.8%)	14 (11.9%)	10 (29.4%)	0.013
SOFA score from day 1 to day 3 on ECMO	9.6 ± 2.3	9.4 ± 2.2	10.6 ± 2.3	0.015
pH from day1 to day 3 on ECMO	7.43 ± 0.08	7.44 ± 0.08	7.42 ± 0.08	0.146
PaCO_2_ (mm Hg) from day1 to day 3 on ECMO	38.2 ± 5.3	38.2 ± 5.1	38.2 ± 6.1	0.999
PaO_2_/FiO_2_ (mm Hg) from day 1 to day 3 on ECMO	178 (131–240)	167 (130–224)	194 (150–248)	0.857
Ventilator settings from day 1 to day 3 on ECMO				
Mechanical power (J/min)	12.1 ± 6.2	12.0 ± 6.2	12.3 ± 6.7	0.867
Tidal volume (mL/kg PBW)	6.0 ± 2.2	6.1 ± 2.1	5.9 ± 2.7	0.705
PEEP (cm H_2_O)	12.0 ± 3.3	12.4 ± 3.4	10.7 ± 2.5	0.008
Peak inspiratory pressure (cm H_2_O)	31.7 ± 5.6	31.6 ± 5.9	32.2 ± 4.8	0.596
Mean airway pressure (cm H_2_O)	17.7 ± 4.0	17.9 ± 4.3	17.0 ± 2.9	0.196
Dynamic compliance (mL/cm H_2_O)	19.2 ± 8.1	19.7 ± 7.7	17.4 ± 9.3	0.153
Minute ventilation (L/min)	5.7 ± 2.8	5.6 ± 2.7	5.8 ± 3.1	0.679
Fluid balance, ml				
Before ECMO	923 (−258 to 2125)	884 (−193 to 2145)	1125 (−486 to 2094)	0.836
Cumulative 3 days	1190 (−873 to 3935)	947 (−1355 to 3253)	1844 (−208 to 5679)	0.046
Inotropes (*n*)	133 (87.5%)	102 (86.4%)	31 (91.2%)	0.568

ARDS: acute respiratory distress syndrome; SOFA: Sequential Organ Failure Assessment; ECMO: extracorporeal membrane oxygenation; PaCO_2_: partial pressure of carbon dioxide in arterial blood; PaO_2_: partial pressure of oxygen in arterial blood; FiO_2_: fraction of inspired oxygen; PBW: predicted body weight; PEEP: positive end-expiratory pressure; Values are expressed as numbers, mean and standard deviation, or median and interquartile range.

**Table 2 membranes-11-00644-t002:** Clinical outcomes of direct and indirect ARDS patients with ECMO.

Outcomes	Direct ARDS	Indirect ARDS	*p*
	(*n* = 118)	(*n* = 34)	
Mortality			
28 day hospital mortality, n (%)	41 (34.7%)	20 (58.8%)	0.005
60 day hospital mortality, n (%)	53 (44.9%)	21 (61.8%)	0.025
90 day hospital mortality, n (%)	56 (47.5%)	21 (61.8%)	0.041
Other outcomes			
Duration of ECMO (days)	10 (6–16)	7 (4–11)	0.044
Duration of mechanical ventilator (days)	22 (12–40)	18 (10–34)	0.201
Length of ICU stay (days)	25 (15–46)	22 (11–38)	0.191
Length of hospital stay (days)	42 (22–65)	33 (19–68)	0.626
ECMO-free days at day 28	0 (0–9)	0 (0–19)	0.126
Ventilator–free days on day 28	0 (0–9)	0 (0–0)	0.135
Ventilator–free days on day 60	0 (0–41)	0 (0–30)	0.145

ARDS: acute respiratory distress syndrome; ECMO: extracorporeal membrane oxygenation; ICU: intensive care unit. Values are expressed as numbers, or median and interquartile range.

**Table 3 membranes-11-00644-t003:** Background characteristics and clinical variables of survivors and non-survivors: Direct and indirect ARDS patients.

Variables	Direct ARDS (*n* = 118)			Indirect ARDS (*n* = 34)		
	Survivors	Non-Survivors	*p*	Survivors	Non-Survivors	*p*
	(*n* = 59)	(*n* = 59)		(*n* = 12)	(*n* = 22)	
Age (years)	47.3 ± 15.7	54.4 ± 16.3	0.017	39.8 ± 19.3	53.2 ± 13.1	0.022
Gender (male)	40 (67.8%)	43 (72.9%)	0.545	8 (66.7%)	12 (54.5%)	0.717
Body mass index (kg/m^2^)	26.4 ± 6.1	25.3 ± 4.7	0.292	24.1 ± 4.2	26.5 ± 5.1	0.218
Diabetes mellitus	21 (35.6%)	12 (20.3%)	0.065	2 (16.7%)	5 (22.7%)	1.000
Chronic liver disease	6 (10.2%)	10 (16.9%)	0.282	0	5 (22.7%)	0.137
Immunocompromised status	8 (13.6%)	20 (33.9%)	0.009	3 (25%)	9 (40.9%)	0.465
Chronic kidney disease	8 (13.6%)	5 (8.5%)	0.378	0	4 (18.2%)	0.273
SOFA score before ECMO	10.4 ± 3.0	10.8 ± 3.1	0.509	9.8 ± 3.2	12.5 ± 3.2	0.024
Lung injury score before ECMO	3.5 ± 0.4	3.3 ± 0.5	0.055	3.2 ± 0.6	3.3 ± 0.4	0.554
ARDS duration before ECMO (h)	10 (5–70)	63 (16–154)	0.003	6 (1–34)	38 (16–355)	0.018
pH before ECMO	7.28 ± 0.13	7.28 ± 0.15	0.731	7.26 ± 0.09	7.26 ± 0.16	0.989
PaCO_2_ (mm Hg) before ECMO	51.6 ± 21.0	53.6 ± 17.4	0.572	46.3 ± 9.9	54.9 ± 21.5	0.117
PaO_2_/FiO_2_ (mm Hg) before ECMO	61 (50–77)	64 (52–107)	0.083	75 (60–101)	57 (51–98)	0.495
Ventilator settings before ECMO						
Mechanical power (J/min)	24.8 ± 10.5	24.6 ± 8.5	0.877	20.5 ± 8.8	20.4 ± 9.8	0.974
Tidal volume (mL/kg PBW)	7.7 ± 2.3	7.8 ± 2.3	0.722	7.6 ± 2.6	7.8 ± 3.1	0.791
PEEP (cm H_2_O)	12.4 ± 2.5	11.9 ± 3.1	0.343	11.3 ± 2.8	11.3 ± 2.8	0.952
Peak inspiratory pressure (cm H_2_O)	33.7 ± 5.9	34.6 ± 7.1	0.481	33.3 ± 7.1	33.2 ± 6.5	0.978
Mean airway pressure (cm H_2_O)	18.7 ± 4.1	19.2 ± 4.7	0.587	16.9 ± 4.3	17.9 ± 4.5	0.554
Dynamic compliance (mL/cm H_2_O)	24.1 ± 12.2	21.4 ± 10.0	0.207	19.9 ± 6.9	23.1 ± 14.1	0.469
Minute ventilation (L/min)	10.9 ± 4.2	11.0 ± 3.5	0.837	9.5 ± 3.1	9.1 ± 3.5	0.740
Ventilatory ratio	2.4 ± 1.0	2.6 ± 1.1	0.280	2.0 ± 0.8	2.4 ± 1.4	0.350
ECMO venoarterial mode	4 (6.8%)	10 (16.9%)	0.153	2 (16.7%)	8 (36.4%)	0.432
SOFA score from day 1 to day 3 on ECMO	8.8 ± 1.8	10.1 ± 2.4	0.001	9.5 ± 2.1	11.3 ± 2.3	0.060
pH from day1 to day 3 on ECMO	7.46 ± 0.06	7.42 ± 0.09	0.006	7.45 ± 0.08	7.40 ± 0.08	0.138
PaCO_2_ (mm Hg) from day1 to day 3 on ECMO	38.2 ± 4.9	38.2 ± 5.3	0.960	37.4 ± 5.5	38.6 ± 6.4	0.586
PaO_2_/FiO_2_ (mm Hg) from day 1 to day 3 on ECMO	194 (145–247)	151 (122–203)	0.844	224 (171–287)	184 (134–237)	0.313
Ventilator settings from day 1 to day 3 on ECMO						
Mechanical power (J/min)	11.2 ± 4.4	12.9 ± 7.4	0.136	9.4 ± 3.4	13.8 ± 7.5	0.026
volume (mL/kg PBW)	6.0 ± 2.0	6.1 ± 2.1	0.878	6.1 ± 2.1	5.8 ± 3.0	0.716
PEEP (cm H_2_O)	12.7 ± 3.2	12.0 ± 3.5	0.299	10.7 ± 2.7	10.7 ± 2.5	0.989
Peak inspiratory pressure (cm H_2_O)	30.8 ± 5.1	32.4 ± 6.5	0.148	29.4 ± 5.4	33.7 ± 3.8	0.011
Mean airway pressure (cm H_2_O)	17.7 ± 3.8	18.1 ± 4.7	0.587	16.1 ± 2.4	17.5 ± 3.1	0.173
Dynamic compliance (mL/cm H_2_O)	21.1 ± 7.6	18.3 ± 7.6	0.055	21.2 ± 8.8	13.6 ± 7.2	0.015
Minute ventilation (L/min)	5.3 ± 2.1	5.9 ± 3.1	0.224	4.9 ± 1.9	6.4 ± 3.5	0.116
Fluid balance, ml						
Before ECMO	991 (−262 to 2278)	795 (−167 to 1723)	0.983	1641 (−753 to 2951)	951 (−486 to 1880)	0.535
Cumulative 3 days	−29 (−1831 to 2108)	2051(−441 to 4534)	0.001	1971 (−708 to 3828)	1803 (77–7267)	0.238
Inotropes (*n*)	43 (72.9%)	59 (100%)	<0.001	10 (83.3%)	21 (95.5%)	0.279

Values are expressed as numbers, mean and standard deviation, or median and interquartile range. ARDS: acute respiratory distress syndrome; SOFA: Sequential Organ Failure Assessment; ECMO: extracorporeal membrane oxygenation; PaCO_2_: partial pressure of carbon dioxide in arterial blood; PaO_2_: partial pressure of oxygen in arterial blood; FiO_2_: fraction of inspired oxygen; PBW: predicted body weight; PEEP: positive end-expiratory pressure.

**Table 4 membranes-11-00644-t004:** Cox proportional hazard regression models for predictors of 90-day hospital mortality.

Variables	Direct ARDS (*n* = 118)		Indirect ARDS (*n* = 34)	
	Adjust HR (95% CI)	*p*	Adjust HR (95% CI)	*p*
Diabetes mellitus	0.246 (0.111–0.546)	0.001		
Immunocompromised status	3.860 (1.943–7.668)	<0.001		
ARDS duration before ECMO (h)	1.002 (1.000–1.004)	0.015		
SOFA score from day 1 to day 3 on ECMO	1.225 (1.070–1.402)	0.003	2.514 (1.474–4.286)	0.001
Dynamic compliance from day 1 to day 3 on ECMO			0.886 (0.802–0.978)	0.017

ARDS: acute respiratory distress syndrome; ECMO: extracorporeal membrane oxygenation; SOFA: sequential organ failure Assessment; HR: hazard ratio; CI: confidence interval. The multivariate analysis models included age, comorbidities, lung injury score before ECMO, ARDS duration before ECMO, ECMO venoarterial mode use, mean sequential organ failure assessment score from day 1 to 3 on ECMO, and mean values of ventilatory variables from day 1 to 3 on ECMO.

**Table 5 membranes-11-00644-t005:** Background characteristics and clinical variables after excluding VA ECMO patients: Direct and indirect ARDS patients.

Variables	All	Direct ARDS	Indirect ARDS	*p*
	(*n* = 128)	(*n* = 104)	(*n* = 24)	
Age (years)	51.2 ± 16.5	51.1 ± 16.6	51.3 ± 16.5	0.975
Gender (male)	89 (69.5%)	72 (69.2%)	17 (70.8%)	0.878
Body mass index (kg/m^2^)	25.8 ± 5.5	25.9 ± 5.6	25.4 ± 4.7	0.708
Diabetes mellitus	38 (29.7%)	31 (29.8%)	7 (29.2%)	0.951
Chronic liver disease	20 (15.6%)	16 (15.4%)	4 (16.7%)	1.000
Immunocompromised status	34 (26.6%)	25 (24%)	9 (37.5%)	0.178
Chronic kidney disease	16 (12.5%)	12 (11.5%)	4 (16.7%)	0.500
SOFA score before ECMO	10.7 ± 3.1	10.5 ± 3.1	11.7 ± 2.8	0.091
Lung injury score before ECMO	3.4 ± 0.4	3.4 ± 0.4	3.2 ± 0.4	0.041
ARDS duration before ECMO (h)	29 (8–133)	29 (8–122)	30 (8–339)	0.143
pH before ECMO	7.28 ± 0.14	7.29 ± 0.14	7.24 ± 0.13	0.186
PaCO_2_ (mm Hg) before ECMO	52.8 ± 19.5	52.7 ± 19.3	53.5 ± 20.6	0.851
PaO_2_/FiO_2_ (mm Hg) before ECMO	63 (52–93)	63 (52–86)	65 (54–107)	0.216
Ventilator settings before ECMO				
Mechanical power (J/min)	24.1 ± 9.8	24.7 ± 9.7	21.5 ± 9.8	0.152
Tidal volume (mL/kg PBW)	7.7 ± 2.3	7.7 ± 2.3	7.8 ± 2.2	0.948
PEEP (cm H_2_O)	12.2 ± 2.7	12.3 ± 2.7	11.5 ± 2.5	0.180
Peak inspiratory pressure (cm H_2_O)	33.9 ± 6.5	34.0 ± 6.2	33.6 ± 7.5	0.772
Mean airway pressure (cm H_2_O)	18.6 ± 4.4	18.8 ± 4.4	17.8 ± 4.7	0.331
Dynamic compliance (mL/cm H_2_O)	22.8 ± 11.3	22.7 ± 11.0	23.1 ± 13.1	0.874
Minute ventilation (L/min)	10.7 ± 3.9	11.0 ± 4.1	9.4 ± 3.1	0.089
Ventilatory ratio	2.5 ± 1.0	2.5 ± 1.0	2.3 ± 1.2	0.426
SOFA score from day 1 to day 3 on ECMO	9.7 ± 2.2	9.5 ± 2.1	11.0 ± 2.1	0.005
pH from day1 to day 3 on ECMO	7.44 ± 0.07	7.44 ± 0.07	7.41 ± 0.08	0.083
PaCO_2_ (mm Hg) from day1 to day 3 on ECMO	38.1 ± 5.0	38.0 ± 4.9	38.6 ± 5.4	0.576
PaO_2_/FiO_2_ (mm Hg) from day 1 to day 3 on ECMO	178 (134–216)	167 (131–214)	194 (156–240)	0.763
Ventilator settings from day 1 to day 3 on ECMO				
Mechanical power (J/min)	11.7 ± 6.1	11.5 ± 6.0	12.2 ± 6.2	0.638
Tidal volume (mL/kg PBW)	5.9 ± 2.0	5.9 ± 2.1	5.6 ± 1.8	0.537
PEEP (cm H_2_O)	12.3 ± 3.3	12.6 ± 3.4	10.8 ± 2.5	0.017
Peak inspiratory pressure (cm H_2_O)	31.3 ± 5.3	31.2 ± 5.3	32.0 ± 5.2	0.467
Mean airway pressure (cm H_2_O)	17.7 ± 3.9	17.9 ± 4.2	17.2 ± 2.7	0.325
Dynamic compliance (mL/cm H_2_O)	19.3 ± 7.9	19.7 ± 7.6	17.6 ± 9.3	0.269
Minute ventilation (L/min)	5.5 ± 2.8	5.4 ± 2.7	5.8 ± 2.9	0.556
Fluid balance, ml				
Before ECMO	938 (−130 to 2125)	938 (−78 to 2204)	957 (−717 to 2002)	0.731
Cumulative 3 days	1190 (−873 to 3846)	801 (−1378 to 3275)	2097 (1100 to 6976)	0.029
Inotropes (*n*)	111 (89.5%)	89 (85.6%)	22 (91.7%)	0.738
28 day hospital mortality, *n* (%)	48 (37.5%)	34 (32.7%)	14 (58.3%)	0.009
60 day hospital mortality, *n* (%)	57 (44.5%)	42 (40.4%)	15 (62.5%)	0.019
90 day hospital mortality, *n* (%)	59 (46.1%)	44 (42.3%)	15 (62.5%)	0.027

VA: venoarterial; ECMO: extracorporeal membrane oxygenation; ARDS: acute respiratory distress syndrome; SOFA: sequential organ failure assessment; PaCO_2_: partial pressure of carbon dioxide in arterial blood; PaO_2_: partial pressure of oxygen in arterial blood; FiO_2_: fraction of inspired oxygen; PBW: predicted body weight; PEEP: positive end-expiratory pressure; Values are expressed as numbers, mean and standard deviation, or median and interquartile range.

## Data Availability

All data will be available from the corresponding author on reasonable request.

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
