# Peer review of "Comparisons of Outcomes between Patients with Direct and Indirect Acute Respiratory Distress Syndrome Receiving Extracorporeal Membrane Oxygenation"

_membranes, 2021, doi:10.3390/membranes11080644_

Round 1
Reviewer 1 Report
- please comment on the diabetes as a protective factor in indirect ARDS
- please comment on the use of corticosteroids if taken into account in your analysis or not . please discuss for relevance citing literature if available
Author Response
Point 1: please comment on the diabetes as a protective factor in indirect ARDS
Response 1: We thank the reviewer’s suggestion to comment on the diabetes as a protective factor in indirect ARDS.
The impact of pre-existing diabetes mellitus on the development of ARDS in critically ill patients remain unclear, and there were conflicting data about whether diabetes as a protective factor to influence clinical outcomes or mortality of ARDS. A retrospective cohort study reported that higher mortality in ARDS patients with diabetes [1], one study showed ARDS patients with diabetes had lower mortality than ARDS patients without diabetes, although not significantly different [2], and another study reported no association of mortality between diabetes and ARDS [3]. However, previous studies evaluating the impact of diabetes on risk of ARDS development, clinical outcomes or mortality of ARDS did not focus on direct or indirect etiology of ARDS.
A previous retrospective observational cohort study was the first study to assess differences in mortality related to diabetes in ARDS by direct and indirect injury [4]. They found that a protective association of diabetes with hospital mortality was only limited to patients with direct ARDS, not in indirect ARDS patients.
There was few study to investigate the impact of diabetes on clinical outcome of ARDS patients undergoing ECMO, and it may be difficult to raise any conclusion about their impact on outcome due to relative small patients [5]. Our report found that diabetes mellitus had an impact on survival in direct ARDS patients receiving ECMO, not in indirect ARDS patients.
We added the above description about diabetes was not found as a protective factor in indirect ARDS in our study in the eighth paragraph of the Discussion section in the revised manuscript as follows (marked with red text):
One retrospective cohort study reported that preexisting diabetes mellitus was independently associated with a reduced risk of mortality only in direct ARDS patients, and diabetes was not found as a protective factor in indirect ARDS patients [11].
Diabetes as a protective factor in indirect ARDS patients undergoing ECMO was not found in our study.
References:
- Koh, G.C.; Vlaar, A.P.; Hofstra, J.J.; de Jong, H.K.; van Nierop, S.; Peacock, S.J.; Wiersinga, W.J.; Schultz, M.J.; Juffermans, N.P. In the critically ill patient, diabetes predicts mortality independent of statin therapy but is not associated with acute lung injury: a cohort study. Crit Care Med. 2012, 40, 1835-43.
- Moss, M.; Guidot, D.M.; Steinberg, K.P.; Duhon, G.F.; Treece, P.; Wolken, R.; Hudson, L.D.; Parsons, P.E. Diabetic patients have a decreased incidence of acute respiratory distress syndrome. Crit Care Med. 2000, 28, 2187-92.
- Yu, S.; Christiani, D.C.; Thompson, B.T.; Bajwa, E.K.; Gong, M.N. Role of diabetes in the development of acute respiratory distress syndrome. Crit Care Med. 2013, 41, 2720-32.
- Luo, L.; Shaver, C.M.; Zhao, Z.; Koyama, T.; Calfee, C.S.; Bastarache, J.A.; Ware, L.B. Clinical Predictors of Hospital Mortality Differ Between Direct and Indirect ARDS. 2017, 151, 755-763.
- Rozencwajg, S.; Pilcher, D.; Combes, A.; Schmidt, M. Outcomes and survival prediction models for severe adult acute respiratory distress syndrome treated with extracorporeal membrane oxygenation. Crit Care. 2016, 20, 392.
Point 2: please comment on the use of corticosteroids if taken into account in your analysis or not . please discuss for relevance citing literature if available
Response 2: We thank the reviewer’s comment to point out this issue, and we apologized for not analysing the effect of corticosteroids on clinical outcomes of severe ARDS patients receiving ECMO in our study due to the retrospective nature.
Corticosteroids have anti-inflammatory and antifibrosis effects although the impact on clinical outcomes in ARDS patients remains controversial, including severe ARDS patients undergoing ECMO. Besides, clinical trials evaluating corticosteroids in the management of ARDS patients showed conflicting results.
A previous multicenter, randomized controlled trial concluded that the routine use of methylprednisolone for persistent ARDS was not suggested, and starting methylprednisolone therapy more than two weeks after the onset of ARDS may increase the risk of death [1].
However, another recent multicenter, randomized controlled trial showed that the early administration of dexamethasone could reduce duration of mechanical ventilation and overall mortality in patients with established moderate-to-severe ARDS [2].
A recent meta-analysis demonstrated that glucocorticoid treatment might reduce overall mortality, duration of mechanical ventilation and improve oxygenation in ARDS patients [3].
However, this meta-analysis did not conduct subgroup analyses such as the underlying etiology of ARDS (i.e., direct or indirect ARDS). This study also found that glucocorticoid treatment was not associated with a higher incidence of new infection and hyperglycemia. However, the optimal dose, timing, type and duration of steroid therapy remains unknown and further studies are needed [3].
Therefore, until today, corticosteroids use in ARDS remains highly controversial due to unclear benefits and optimal dose, timing, class, and duration of corticosteroids remain unknown.
All the above studies did not focus on the effect of corticosteroids in direct or indirect ARDS patients respectively although corticosteroids remains highly controversial in overall ARDS population, and may be harmful in influenza related ARDS, i.e., administration of corticosteroids in patients with severe influenza pneumonia is associated with increased ICU mortality [4].
The study of corticosteroids in severe ARDS patients undergoing ECMO was scarce and need further research in the future. We apologized again for not analysing the possible effect of corticosteroids on clinical outcomes of severe ARDS patients receiving ECMO.
We added this limitation in the ninth paragraph of the Discussion section in the revised manuscript as follows (marked with red text):
Fourth, although a recent meta-analysis demonstrated that corticosteroids treatment might reduce overall mortality and duration of mechanical ventilation in ARDS patients [34], they did not conduct subgroup analyses such as the underlying etiology of ARDS (i.e., direct or indirect injury). Corticosteroids use in ARDS remains highly controversial due to unclear benefits and optimal dose and duration are unknown, and our study did not evaluate the possible impact of corticosteroids on clinical outcomes in ARDS patients receiving ECMO.
We also added the reference 34 in the revised manuscript.
References:
- Steinberg, K.P.; Hudson, L.D.; Goodman, R.B.; Hough, C.L.; Lanken, P.N.; Hyzy, R.; Thompson, B.T.; Ancukiewicz, M. National Heart, Lung, and Blood Institute Acute Respiratory Distress Syndrome (ARDS) Clinical Trials Network. Efficacy and safety of corticosteroids for persistent acute respiratory distress syndrome. N Engl J Med. 2006, 354, 1671-84.
- Villar, J.; Ferrando, C.; Martínez, D.; Ambrós, A.; Muñoz, T.; Soler, J.A.; Aguilar, G.; Alba, F.; González-Higueras, E.; Conesa, L.A.; et al. dexamethasone in ARDS network. Dexamethasone treatment for the acute respiratory distress syndrome: a multicentre, randomised controlled trial. Lancet Respir Med. 2020, 8, 267-276.
- Lin, P.; Zhao, Y.; Li, X.; Jiang, F.; Liang, Z. Decreased mortality in acute respiratory distress syndrome patients treated with corticosteroids: an updated meta-analysis of randomized clinical trials with trial sequential analysis. Crit Care. 2021, 25, 122.
- Moreno, G.; Rodríguez, A.; Reyes, L.F.; Gomez, J.; Sole-Violan, J.; Díaz, E.; Bodí, M.; Trefler, S.; Guardiola, J.; Yébenes, J.C.; et al. GETGAG Study Group. Corticosteroid treatment in critically ill patients with severe influenza pneumonia: a propensity score matching study. Intensive Care Med. 2018, 44, 1470-1482.
We thank the reviewer for valuable comments. Addressing them fully has significantly strengthened the manuscript.

Reviewer 2 Report
Dear editor, thank you for giving me the opportunity to revise this interesting paper on the different features of direct vs indirect ARDS in terms of clinical presentation, predictors of outcome and mortality. The topic is interesting to the clinical community; yet, the work is flawed by major limitations.
I see that
- the enrollment period is very long, before even the Berlin definition was released
- There are VA ECMO supported patients which should be excluded from analysis
- No external validation is performed and therefore the numbers are limited and not reproducible (as indeed, stated by the authors themselves)
- Too much retrospective data and calculations are included for analysis
Author Response
Point 1: the enrollment period is very long, before even the Berlin definition was released
Response 1: We thank the reviewer to point out this problem. The enrollment period of our study is very long (2006-2015), even before the Berlin definition in 2012. Although the ECMO techniques were not different in our hospital during the long study period (2006-2015), and a unified ECMO system (most device are Terumo model) was used
We added this limitation in the ninth paragraph of the Discussion section in the revised manuscript as follows (marked with red text):
…the fact that all patients were from the long enrollment period of the previous years (2006–2015) not the recent years in a single tertiary care referral center.
Reference:
- Ranieri, V.M.; Rubenfeld, G.D.; Thompson, B.T.; Ferguson, N.D.; Caldwell, E.; Fan, E.; Camporota, L.; Slutsky, A.S. Acute respiratory distress syndrome: The Berlin Definition. JAMA 2012, 307, 2526-2533.
Point 2: There are VA ECMO supported patients which should be excluded from analysis
Response 2: We thank the reviewer to point out this issue and suggestion. We agreed with the reviewer that VA ECMO and VV ECMO are different techniques and VA ECMO should be excluded from analysis.
VV ECMO was consider for isolated respiratory failure in severe ARDS patients with refractory hypoxemia, and VA ECMO is less frequently applied unless there is significant cardiac dysfunction requiring hemodynamic support, such as unstable hemodynamic compromise, cardiogenic shock, cardiopulmonary cerebral resuscitation, myocardial poor performance or pulmonary hypertension.
We excluded the 24 VA ECMO supported patients, and added Table 5 in the revised manuscript. The primary aim of our study to investigate the correlations among clinical and ventilatory variables, clinical outcomes and mortality in patients with direct or indirect ARDS undergoing ECMO. After excluding 24 VA ECMO supported patients, SOFA score and cumulative fluid balance during the first 3 days of ECMO were still significantly higher in indirect ARDS patients. The 28-, 60-, and 90-day hospital mortality rates were also still significantly higher among patients with indirect ARDS than among patients with direct ARDS (all p < 0.05).
We added paragraph to compare the differences between direct and indirect ARDS patients in the sixth paragraph of the Results section in the revised manuscript after excluding VA ECMO patients (3.6. Comparisons of Direct and Indirect ARDS Patients after Excluding VA ECMO Patients) as follows (marked with red text):
As shown in Table 5, there were no significant difference between direct and indirect ARDS patients in terms of age, gender, body mass index, or comorbidities. Before ECMO initiation, direct ARDS patients had significantly higher lung injury scores. In terms of ventilator settings prior to ECMO, no significant difference between two groups were found.
After ECMO initiation, no significant differences were observed between the two groups in terms of ventilator settings, except for higher PEEP in direct ARDS patients. SOFA scores and cumulative fluid balance were significantly higher in indirect ARDS patients during the first 3 days of ECMO (both p < 0.05). The 28-, 60-, and 90-day hospital mortality rates were significantly higher among indirect ARDS patients than among direct ARDS patients (all p < 0.05).
We added the statement to describe that mortality was still significantly higher in indirect ARDS patients after excluding VA ECMO patients in the fifth paragraph of the Discussion section in the revised manuscript as follows (marked with red text):
However, after excluding VA ECMO supported patients, the 28-, 60-, and 90-day hospital mortality rates were still significantly higher among patients with indirect ARDS.
Point 3: No external validation is performed and therefore the numbers are limited and not reproducible (as indeed, stated by the authors themselves)
Response 3: We appreciate with the reviewer’s comment to point out this problem. Our data is collected retrospectively from a tertiary care referral center, patients in our institution have more comorbidities with higher mortality, and therefore it may provide considerable problems to make external validation of our findings to other ECMO cohorts. We agreed with the reviewer’s comment entirely that this make the numbers in our study limited and not reproducible.
We added this limitation of not performing exertional validation in the ninth paragraph of the Discussion section in the revised manuscript as follows (marked with red text):
First, a relatively small number of indirect ARDS patients, the retrospective nature of our analysis, external validation not performed, and the fact that all patients were from the long enrollment period of the previous years (2006–2015) not the recent years in a single tertiary care referral center.
Point 4: Too much retrospective data and calculations are included for analysis
Response 4: We thank the reviewer’s comment to point out these problems.
Randomized controlled trials of ECMO for severe ARDS comprise a number of uncertainty with ethical and methodological issues like life-saving support equipment in other acute fatal illness.Therefore, prospective randomized trials of ECMO may not be easily to perform.
Another issue is possible high crossover from the control group to ECMO treatment. The ECMO to Rescue Lung Injury in Severe ARDS (EOLIA) trial was stopped early, and 28% of the patients in the control group crossed over to the ECMO group [1].
Therefore, we performed this retrospective study. We agreed with the reviewer that too much retrospective data, and we had mentioned in the limitation in the ninth paragraph of the Discussion section in the revised manuscript.
The aim in this study was to examine correlations among clinical and ventilatory variables in patients with direct or indirect ARDS undergoing ECMO, like mechanical power and ventilatory ratio, and some calculations are included for analysis.
Reference:
- Combes, A.; Hajage, D.; Capellier, G.; Demoule, A.; Lavoué, S.; Guervilly, C.; Da, Silva. D.; Zafrani, L.; Tirot, P.; Veber, B.; EOLIA Trial Group, REVA, and ECMONet. Extracorporeal Membrane Oxygenation for Severe Acute Respiratory Distress Syndrome. N Engl J Med. 2018, 378, 1965-1975.
We thank the reviewer for valuable comments. Addressing them fully has significantly strengthened the manuscript.

Reviewer 3 Report
Dear Authors,
that's a nice retrospective study on the outcomes of pulmonary and extrapulmonary ARDS. Despite not novel it is worth to read data on the topic.
I have some concerns:
- generally in title abstract and study i would define pulmonary and extrapulmonary ards
- the term subphenotype starts to have a precise significance: markers or precision medicine
- i would suggest to explore more and new markers to follow the disease like microrna
- for extrapulmonary ARDS you could discuss the fact thet mortality may be higher due to complications e.g. neurological and abdominal. and the difficulties in treating these complications. would you be able to find data on neurological bleeding or abdominal complications in trauma and pancreatitis?
- I would suggest to discuss more the very interesting data that you have about the fluid balance, probably one major determinant of mortality. and reasonable giving the cause of ARDS
Author Response
Point 1: 
 generally in title abstract and study i would define pulmonary and extrapulmonary ards
Response 1: We thank the reviewer’s suggestion to define pulmonary (direct) and extrapulmonary (indirect) ARDS in abstract and study.
The risk factors of ARDS are divided into direct and indirect causes of lung injury. Direct (pulmonary) ARDS results from an insult that directly affects lung parenchyma (e.g., pneumonia, aspiration of gastric contents), and indirect (extrapulmonary) ARDS results from extrapulmonary sites with an acute systemic inflammatory response (e.g., nonpulmonary sepsis, trauma, pancreatitis).
We added the description to define pulmonary and extrapulmonary ARDS in the abstract as follows (marked with red text):
Acute respiratory distress syndrome (ARDS) is a heterogeneous syndrome caused by direct (local damage to lung parenchyma) or indirect lung injury (insults from extrapulmonary sites with acute systemic inflammatory response).
We added the description to define pulmonary and extrapulmonary ARDS in the second paragraph of the Introduction section in the revised manuscript as follows (marked with red text):
Direct (primary or pulmonary) ARDS results from an insult that directly affects lung parenchyma (e.g., pneumonia, aspiration of gastric contents), and indirect (secondary or extrapulmonary) ARDS results from an insult outside of the lungs with an acute systemic inflammatory response (e.g., nonpulmonary sepsis, trauma, pancreatitis).
Point 2: the term subphenotype starts to have a precise significance: markers or precision medicine
Response 2: We thank the reviewer’s comment and agreed that subphenotype of diseases have a precise significance, including ARDS.
Identifying phenotypes of ARDS based on clinical, physiologic, radiographic, microbiologic and biologic variables and integrating this information to select patients for clinical trials may increase the chance for efficacy with new treatments [1, 2].
Identification of ARDS subphenotypes may help the precision critical care for ARDS [3, 4]. However, personalized approaches to ARDS trials or treatment will need to consider the timing of the intervention, as the clinical and biological phenotype of ARDS evolves rapidly over the first few hours to days of illness [2].
We added the statement about identification of ARDS subphenotypes help precision medicine for management of ARDS in the second paragraph of the Introduction section in the revised manuscript as follows (marked with red text):
..and identification of ARDS subphenotypes may enable the aspiration of precision medicine for ARDS.
References:
- Beitler, J.R.; Thompson, B.T.; Baron, R.M.; Bastarache, J.A.; Denlinger, L.C.; Esserman, L.; Gong, M.N.; LaVange, L.M.; Lewis, R.J.; Marshall, J.C.; et al. Advancing precision medicine for acute respiratory distress syndrome. Lancet Respir Med. 2021, S2213-2600(21)00157-0. Epub ahead of print.
- Matthay, M.A.; Arabi, Y.M.; Siegel, E.R.; Ware, L.B.; Bos, L.D.J.; Sinha, P.; Beitler, J.R.; Wick, K.D.; Curley, M.A.Q.; Constantin, J.M.; et al. Phenotypes and personalized medicine in the acute respiratory distress syndrome. Intensive Care Med. 2020, 46, 2136-2152.
- Sinha, P.; Calfee, C.S. Phenotypes in acute respiratory distress syndrome: moving towards precision medicine. Curr Opin Crit Care. 2019, 25, 12-20.
- Wilson, J.G.; Calfee, C.S. ARDS Subphenotypes: Understanding a Heterogeneous Syndrome. Crit Care. 2020, 24, 102.
Point 3: i would suggest to explore more and new markers to follow the disease like microrna
Response 3: We thank the reviewer’s recommendation to explore more and new markers to follow the disease, like microRNA. MicroRNAs are endogenous single-stranded non-coding small RNA molecules that can be secreted into the circulation and exist stably, and circulating microRNAs exhibited promising potential to serve as effective non-invasive cancer biomarkers for clinical application [1]. A recent study showed that circulating miRNAs could represent promising biomarkers to monitor the evolution of ARDS under ECMO support [2].
Biomarkers are being proposed as measures to identify subphenotypes of ARDS [3]. There were some study reported biologic and clinical makers to follow up ARDS and predict survival, and the biomarker-based risk model reliably identifies ARDS subjects at risk of death [4, 5]. Previous studies used clinical parameters to follow up disease and predict the outcomes of ARDS patients receiving ECMO as predictive survival models [6], like the PRESERVE score [7] and the RESP score [8].
Despite the developed clinical and biologic markers for ARDS and predictive survival models for ARDS patients with ECMO, there was not clearly definite biologic or clinical markers in ARDS that could predict outcomes or mortality precisely, and the morbidity and mortality of ARDS remained high. We agreed with the reviewer that exploring more and new markers to follow up disease could better predict clinical outcomes.
Because our current study is a retrospective analysis, biological (e.g., inflammatory cytokines or biomarkers of lung epithelial injury or endothelial injury), physiologic, radiographic, and some clinical markers were not checked and analysed.
We apologized again for not exploring more and new markers and focused only on ARDS subphenotype of direct or indirect lung injury to follow up the disease due to the retrospective nature, and we will explore more and new makers in our next study in the future.
We added the statement as limitation in the ninth paragraph of the Discussion section in the revised manuscript as follows (marked with red text):
…and explore more and new markers to follow up disease may need further research in the future.
References:
- Wang, H.; Peng, R.; Wang, J.; Qin, Z.; Xue, L. Circulating microRNAs as potential cancer biomarkers: the advantage and disadvantage. Clin Epigenetics. 2018, 10, 59.
- Martucci, G.; Arcadipane, A.; Tuzzolino, F.; Occhipinti, G.; Panarello, G.; Carcione, C.; Bertani, A.; Conaldi, P.G.; Miceli, V. Circulating miRNAs as Promising Biomarkers to Evaluate ECMO Treatment Responses in ARDS Patients. Membranes 2021, 11, 551.
- Sinha, P.; Calfee, C.S. Phenotypes in acute respiratory distress syndrome: moving towards precision medicine. Curr Opin Crit Care. 2019, 25, 12-20.
- Bime, C.; Casanova, N.; Oita, R.C.; Ndukum, J.; Lynn, H.; Camp, S.M.; Lussier, Y.; Abraham, I.; Carter, D.; Miller, E.J.; et al. Development of a biomarker mortality risk model in acute respiratory distress syndrome. Crit Care. 2019, 23, 410.
- Matthay, M.A.; Arabi, Y.M.; Siegel, E.R.; Ware, L.B.; Bos, L.D.J.; Sinha, P.; Beitler, J.R.; Wick, K.D.; Curley, M.A.Q.; Constantin, J.M.; et al. Phenotypes and personalized medicine in the acute respiratory distress syndrome. Intensive Care Med. 2020, 46, 2136-2152.
- Rozencwajg, S.; Pilcher, D.; Combes, A.; Schmidt, M. Outcomes and survival prediction models for severe adult acute respiratory distress syndrome treated with extracorporeal membrane oxygenation. Crit Care. 2016, 20, 392.
- Schmidt, M.; Zogheib, E.; Rozé, H.; Repesse, X.; Lebreton, G.; Luyt, C.E.; Trouillet, J.L.; Bréchot, N.; Nieszkowska, A.; Dupont, H.; et al. The PRESERVE mortality risk score and analysis of long-term outcomes after extracorporeal membrane oxygenation for severe acute respiratory distress syndrome. Intensive Care Med. 2013, 39, 1704–1713.
- Schmidt, M.; Bailey, M.; Sheldrake, J.; Hodgson, C.; Aubron, C.; Rycus, P.T.; Scheinkestel, C.; Cooper, D.J.; Brodie, D.; Pellegrino, V. Predicting survival after extracorporeal membrane oxygenation for severe acute respiratory failure. The Respiratory Extracorporeal Membrane Oxygenation Survival Prediction (RESP) score. Am J Respir Crit Care Med. 2014, 189, 1374-82.
Point 4: for extrapulmonary ARDS you could discuss the fact thet mortality may be higher due to complications e.g. neurological and abdominal. and the difficulties in treating these complications. would you be able to find data on neurological bleeding or abdominal complications in trauma and pancreatitis?
Response 4: We thank the reviewer’s comment to point out this issue, and we agreed that the complications of underlying diseases may contribute to higher mortality in extrapulmonary ARDS. There were 6 patients had trauma and 4 patients had acute pancreatitis as the risk factors of ARDS in our study. Among 6 ARDS patients due to trauma, one patient died due to intracranial haemorrhage. Among 4 ARDS patients due to acute pancreatitis, two patients died due to necrotizing pancreatitis.
We added the description to discuss the fact that mortality may be higher due to complications of underlying diseases in extrapulmonary ARDS in the second paragraph of the Discussions section in the revised manuscript as follows (marked with red text):
The higher mortality in indirect (extrapulmonary) ARDS patients may be due to the complications of underlying diseases and the difficulties in treating these fatal complications [19]. Our study showed that one ARDS patient due to trauma had intracranial hemorrhage and two ARDS patients due to acute pancreatitis had necrotizing pancreatitis, and these complications eventually contributed to mortality.
We also added the reference 19 in the revised manuscript.
References:
- Wang, C.; Zhang, L.; Qin, T.; Xi, Z.; Sun, L.; Wu, H.; Li, D. Extracorporeal membrane oxygenation in trauma patients: a systematic review. World J Emerg Surg. 2020, 15, 51.
- Robba, C.; Ortu, A.; Bilotta, F.; Lombardo, A.; Sekhon, M.S.; Gallo, F.; Matta, B.F. Extracorporeal membrane oxygenation for adult respiratory distress syndrome in trauma patients: A case series and systematic literature review. J Trauma Acute Care Surg. 2017, 82, 165-173.
Point 5: I would suggest to discuss more the very interesting data that you have about the fluid balance, probably one major determinant of mortality. and reasonable giving the cause of ARDS
Response 5: This is an excellent point of view. We appreciated the reviewer’s comment and agreed that fluid balance during early phase of ECMO may be one major determinant of mortality in severe ARDS patients.
Previous studies reported that early positive fluid balance was independently associated with mortality in patients undergoing ECMO support [1, 2]. Sepsis is the main risk factor of indirect ARDS. Indirect ARDS patients had higher risk of hemodynamic impairment or shock, which may be due to higher percentage of sepsis, and therefore a higher proportion of these patients may receive more fluid resuscitation and vasopressors [3]. Our findings showed that there was no significant difference in SOFA score and fluid balance before ECMO between direct and indirect ARDS patients. However, during the first 3 days of ECMO, SOFA scores and cumulative fluid balance were significantly higher in the indirect ARDS group (both p < 0.05).
The most common cause of death among ARDS patients is multiorgan failure. Excess fluid accumulation may exacerbate tissue edema, stretch the vascular wall, worsen vascular permeability, and ultimately develop organ dysfunction with corresponding effects on clinical outcomes and mortality [4]. The causal relationship between fluid overload and organ dysfunction was difficult to be determined due to the retrospective nature of our study. Indirect ARDS patients in our study had higher percentage of sepsis, which may have higher percentage of hemodynamic impairment requiring more fluid resuscitation and vasopressors during early phase of ECMO. Indirect ARDS patients had significantly higher cumulative fluid balance and higher organ failure than direct ARDS patients, which may contribute to higher mortality.
We added the above description about fluid balance in severe ARDS undergoing ECMO in the sixth paragraph of the Discussion section in the revised manuscript as follows (marked with red text):
During the early phase of ECMO, positive fluid balance was independently associated with mortality [27]. Excess fluid accumulation may exacerbate tissue edema, stretch the vascular wall, worsen vascular permeability, and ultimately develop organ dysfunction with corresponding effects on clinical outcomes and mortality [28]. Although the causal relationship between fluid overload and organ dysfunction was difficult to be determined due to the retrospective nature of our study, our findings showed that indirect ARDS patients had significantly higher cumulative fluid balance and higher organ failure during early phase of ECMO than direct ARDS patients, which may contribute to higher mortality.
We also added the reference 27 and 28 in the revised manuscript.
References:
- Schmidt, ; Bailey, M.; Kelly, J.; Hodgson, C.; Cooper, D.J.; Scheinkestel, C.; Pellegrino, V.; Bellomo, R.; Pilcher, D. Impact of fluid balance on outcome of adult patients treated with extracorporeal membrane oxygenation. Intensive Care Med. 2014, 40, 1256–1266.
- Kim, ; Paek, J.H.; Song, J.H.; Lee, H.; Jhee, J.H.; Park, S.; Yun, H.R.; Kee, Y.K.; Han, S.H.; Yoo, T.H.; et al. Permissive fluid volume in adult patients undergoing extracorporeal membrane oxygenation treatment. Crit. Care 2018, 22, 270.
- Luo, L.; Shaver, C.M.; Zhao, Z.; Koyama, T.; Calfee, C.S.; Bastarache, J.A.; Ware, L.B. Clinical Predictors of Hospital Mortality Differ Between Direct and Indirect ARDS. 2017, 151, 755-763.
- Ostermann, ; Straaten, H.M.; Forni, L.G. Fluid overload and acute kidney injury: Cause or consequence? Crit. Care 2015, 19, 443.
We thank the reviewer for valuable comments. Addressing them fully has significantly strengthened the manuscript.

Reviewer 4 Report
Dear authors
I read with interest your manuscript.
Overall it is well written and sufficient informative.
I suggest to the editor to publish it.
MY-mayor comment:
Why you do not use primary and secondary ARDS?
The case is from a previous year and not recent.
Best Regards
Author Response
Point 1: Why you do not use primary and secondary ARDS?
Response 1: We thank the reviewer’s comment. Primary and secondary ARDS also represented direct and indirect ARDS respectively.
The risk factors of ARDS are divided into direct and indirect causes of lung injury. Direct (primary or pulmonary) ARDS results from an insult that directly affects lung parenchyma (e.g., pneumonia, aspiration of gastric contents), and indirect (secondary or extrapulmonary) ARDS results from an acute systemic inflammatory response (e.g., nonpulmonary sepsis, trauma, pancreatitis).
Therefore, primary ARDS represents lung injury from a direct insult to lung parenchyma, i.e., direct or pulmonary ARDS. Secondary ARDS represents lung injury from systemic process, i.e., indirect or extrapulmonary ARDS.
We added some description to define primary and secondary ARDS in the second and third paragraph of the Introduction section in the revised manuscript, and added the words “primary” and “secondary” ARDS (marked with red text).
Point 2: The case is from a previous year and not recent.
Response 2: We thank the reviewer to point out this problem that this study enrolled severe ARDS patients receiving ECMO from a previous year (2006-2015), i.e., not during the recent years. Although the ECMO techniques were not different in our hospital between the study period (2006-2015) and the recent years, and a unified ECMO system (most device are Terumo model) was used during the study period and the recent years.
We added this limitation in the ninth paragraph of the Discussion section in the revised manuscript as follows (marked with red text):
…and the fact that all patients were from the long enrollment period of the previous years (2006–2015) not the recent years in a single tertiary care referral center.
We thank the reviewer for valuable comments. Addressing them fully has significantly strengthened the manuscript.

Round 2
Reviewer 2 Report
The authors have thoroughly addressed all the points raised by the reviewer.
Reviewer 3 Report
Thanks for following the suggestions